

# A diagnostic model for overweight and obesity from untargeted urine metabolomics of soldiers

Exsal M. Albores-Mendez[1], Alexis D. Aguilera Hernández[1], Alejandra Melo-González[1], Marco A. Vargas-Hernández[1], Neptalí Gutierrez de la Cruz[1], Miguel A. Vazquez-Guzman[1,2], Melchor Castro-Marín[1], Pablo Romero-Morelos[1,3] and Robert Winkler[4,5]

[1] Escuela Militar de Graduados de Sanidad, Universidad del Ejército y Fuerza Aérea Mexicanos, Secretaría de la Defensa Nacional, Mexico City, Mexico
[2] Centro de Investigación en Ciencias de la Salud (CICSA), FCS, Universidad Anahuac Mexico, Campus Norte, Mexico City, Mexico
[3] Universidad Estatal del Valle de Ecatepec, Ecatepec, Mexico
[4] UGA-Langebio, CINVESTAV, Irapuato, Gto., Mexico
[5] Biotechnology and Biochemistry, CINVESTAV Unidad Irapuato, Irapuato, Gto., Mexico

Corresponding author
Robert Winkler,
robert.winkler@cinvestav.mx

## ABSTRACT

Soldiers in active military service need optimal physical fitness for successfully carrying out their operations. Therefore, their health status is regularly checked by army doctors. These inspections include physical parameters such as the body-mass index (BMI), functional tests, and biochemical studies. If a medical exam reveals an individual's excess weight, further examinations are made, and corrective actions for weight lowering are initiated. The collection of urine is non-invasive and therefore attractive for frequent metabolic screening. We compared the chemical profiles of urinary samples of 146 normal weight, excess weight, and obese soldiers of the Mexican Army, using untargeted metabolomics with liquid chromatography coupled to high-resolution mass spectrometry (LC-MS). In combination with data mining, statistical and metabolic pathway analyses suggest increased S-adenosyl-L-methionine (SAM) levels and changes of amino acid metabolites as important variables for overfeeding. We will use these potential biomarkers for the ongoing metabolic monitoring of soldiers in active service. In addition, after validation of our results, we will develop biochemical screening tests that are also suitable for civil applications.

## INTRODUCTION

Many professionals require a certain level of physical fitness for their work, particularly first-line responders such as firefighters, paramedics, and military personnel. To ensure their operability, they require, in addition to training, good eating habits and periodic review of their health status.

Overweight and obesity are present in most populations and are the origin of numerous metabolic diseases (*Kaplan, 1989*; *Tchernof & Després, 2013*; *Cirulli et al., 2019*). The World Health Organization (WHO) recognizes obesity as a global epidemic (*James, 2008*).

In Mexico, the prevalence of overweight and obesity is dramatically high at about 75% (*Instituto Nacional de Salud Pública (MX), 2018*). Thus, the Mexican official standard NOM-008-SSA3-2010 for the comprehensive management of obesity defines obesity as a public health problem in Mexico due to its magnitude and impact. Criteria for health management should support the early detection, prevention, comprehensive treatment, and control of the growing number of patients (*Secretaría de Gobernación (MX), 2010*).

Soldiers of the Mexican Army have regular exams of their health state by a military doctor. Since overweight and obese soldiers could present risks for their own health and missions, mainly in the special bodies such as paratroopers, they are sent to lose weight in particular training camps such as the "Center for improving lifestyle and health" in Mexico City. Furthermore, the social security institute's law for the Mexican Armed Forces considers soldiers with a Body Mass Index (BMI) greater than 30 as incapable of active service (*Cámara de Diputados (MX), 2019* ). This medical assessment of the soldiers measures vital signs, weight, height, calculating the BMI, clinical history, and a meticulous clinical examination of the body's apparatus and systems. Additional laboratory and cabinet studies are indicated if the doctor identifies alterations or abnormalities in these clinical analyses. All these studies could reveal possible diseases. However, for the case of overweight and obesity, the diagnosis is currently only based on the calculation of the BMI without considering important aspects such as the patient's physiological and metabolic status.

Metabolites in body fluids can be analyzed to assess the nutrition and endogenous changes associated with overweight and obesity, using techniques such as nuclear magnetic resonance (NMR) and mass spectrometry (MS) (*Xie, Waters & Schirra, 2012*; *Zhang, Sun & Wang, 2013*). Usually, invasive studies such as blood analyses explore the patients' metabolic changes and monitor corrective actions. On the other hand, non-invasive tests are generally limited to phenotypic measurements such as body mass index.

Analyzing urine would be more convenient for patients and provide information on the metabolism and pathways involved in particular conditions (*Braga, 2017*). Urine is a biofluid that contains different molecules generated by the organism's metabolism that must be eliminated and represents an excellent source of human sample material because it is available non-invasively. Typically, various molecules are altered simultaneously in diseased people (*Bruzzone et al., 2021*).

Artificial intelligence and machine learning algorithms can support medical diagnosis (*Hatwell, Gaber & Azad, 2020*). Classification is the most widely implemented machine learning task in the medical sector, employing, for example, the Adaptive Boost algorithm (*Freund, 2001*). Adaptive Boost pre-processing also helps to select the most important features automatically from high dimensional data and decision trees (*Rangini & Jiji, 2013*).

This study used untargeted metabolomics based on mass spectrometry to analyze urine from military personnel with normal and excess weight (overweight and obesity). Using

Ada Boost data mining, we created a classification model and identified possible biomarkers for monitoring the metabolic state of soldiers and the early diagnosis of deviations.

## MATERIALS AND METHODS

### Participants and sample preparation

Participants were recruited from the Military Medical Sciences Center, Mexico City, Mexico. Inclusion criteria were: both sexes, active military service, and signed consent to participate voluntarily. Participants answered a questionnaire to identify risk factors for obesity; the next day, nutritional status was assessed by bioelectrical impedance.

The Body-Mass-Index (BMI) was calculated using Eq. (1), according to the WHO definition (*World Health Organization (WHO), 2021*):

$$BMI = \frac{mass}{height^2} \tag{1}$$

with the person's weight measured in kilograms (kg) and the person's height in meters (m).

Following the WHO system, soldiers with a BMI equal to or higher than 25 were classified as 'overweight,' and those with a BMI equal to or above 30 as 'obese' (*World Health Organization (WHO), 2021*).

The first urine of the day was collected at 6 am, and the samples were frozen at −60 °C until their processing. Urine samples were thawed and centrifuged at 850 g for 5 min for metabolomics analysis. Ten L of each sample were diluted in 90 L of chromatography-mass spectrometry (LC-MS) grade water (1:9 *v/v*) and transferred to vials for UPLC-MS analysis.

### Untargeted metabolomics by HPLC-MS

LC-MS grade acetonitrile, water, and acetic acid were purchased from JT Baker (Brick Town, NJ, USA). Samples were analyzed with a Dionex UltiMate 3000 HPLC (Thermo Scientific, Waltham, MA, USA) coupled to an Orbitrap Fusion Tribrid Mass Spectrometer (Thermo Scientific) with an electrospray ionization source. We used an AccuCore C18 column (4.6 × 150 mm, 2.6 m) to separate metabolites using a binary gradient elution of solvents A and B, similar to the method described by *López-Hernández et al. (2019)*. In short, the mobile phase was A: 0.5% acetic acid in water; B: 0.5% acetic acid in acetonitrile. The mobile phase was delivered at a flow rate of 0.5 mL/min, initially with 1% B, followed by a linear gradient to 15% B over 3 min. Solvent B was increased to 50% within 3 min. Over the next 4 min, the gradient was ramped up to 90% B with a plateau for 2 min. The amount of B was then decreased to 50% in 2 min. 2 min later, the solvent B was lowered to 15%, and finally, solvent B returned to initial conditions(1%) until the end of the chromatographic run (18 min). The column temperature was controlled at 40 °C. The injection volume was 20 L.

Data were acquired in positive electrospray ionization (ESI+) mode with the capillary voltage set to 3.5 kV, the Ion Transfer Tube Temperature to 350 °C, and Vaporizer Temp to 400 °C. The desolvation gas was nitrogen with a flow rate of 50 UA (arbitrary units). The detector type was Orbitrap at a resolution of 120,000. Data were acquired from

50–2,000 *m/z* in Full Scan mode with an AGC target of 2.0E5. Before the analysis, the mass spectrometer was calibrated with LTQ ESI Positive Ion Calibration Solution (Pierce, Thermo Scientific).

## Conversion of raw files to mzML

We used the docker version of the ProteoWizard `msconvert` tool (https://proteowizard.sourceforge.io/) (*Kessner et al., 2008*). To reduce disk space and memory use during file processing, we downsampled the data to 32-bit, peak picking, and `zlib` compression:

```
> docker run -it --privileged=true -v /home/rob/dataspace/SUPEREGO/
raw_data/:/data
chambm/pwiz-skyline-i-agree-to-the-vendor-licenses bash

root@0926785f04fc:/data# wine msconvert *.raw --32 --zlib --filter
"peakPicking true 1-" --filter "zeroSamples removeExtra"
```

## Processing of mzML files with KNIME

For mass spectrometry raw data processing and generation of an aligned feature matrix, we employed the OpenMS nodes (*Sturm et al., 2008*; *Pfeuffer et al., 2017*; *Röst et al., 2016*) of the KNIME Analytics Platform (https://www.knime.com) (*Berthold et al., 2009*; *Alka et al., 2020*). Figure 1 represents the KNIME workflow for the raw data processing and matrix generation. The exact parameters of each step are documented in the `workflow.knime` workflow file, provided as Supplementary Files at Zenodo (see 'Data Availability' statement below). For preparing the resulting table of aligned features for the MetaboAnalyst Web Server (*Xia et al., 2009*), we edited the `.CSV` file with `vim` (https://www.vim.org/), using the CSV vim plugin (<chrisbra/csv.vim>).

## Statistical analyses with MetaboAnalyst

For metabolic classification models, we used the web-based version of MetaboAnalyst (https://www.metaboanalyst.ca/) (*Xia et al., 2009*; *Chong, Yamamoto & Xia, 2019*; *Wishart, 2020*). We applied the one-factor statistical analysis for peak intensities in a plain text file, with unpaired samples in columns.

The MetaboAnalyst report for the uploaded data is provided as a Supplemental File.

First, we filtered the raw data by the interquartile range (IQR), normalized it by the median, and applied a square root transformation. Further, we used auto-scaling, *i.e.,* the values were mean-centered and divided by the standard deviation of each variable.

## Metabolic pathway enrichment and metabolite identification

For identifying metabolic pathway enrichment and likely involved metabolites, we used the Functional Analysis (MS peaks) tool of MetaboAnalyst (*Li et al., 2013*). We specified a mass search against the Human Metabolome Database (HMDB, https://hmdb.ca) (*Wishart et al., 2018*; *Wishart et al., 2022*), with 10 ppm mass tolerance in positive mode. We filtered raw data by the interquartile range (IQR), normalized by the median, and applied a square root transformation. Further, we used auto-scaling, *i.e.,* the values were mean-centered

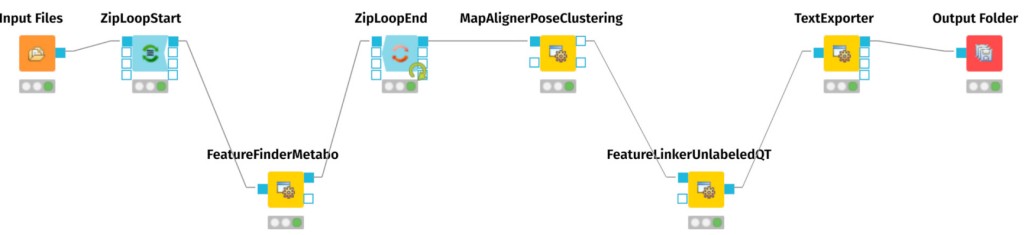

**Figure 1** **KMIME-Workflow for processing the urinary metabolomics data.** The final result is an aligned matrix of features.

and divided by the standard deviation of each variable (the same data preparation as for statistics above). For the Mummichog algorithm, we set a *p*-value cutoff of 0.25 (default top: 10% peaks). We used the pathway library of *Homo sapiens* MFN pathway/metabolite sets (a meta library) with at least five entries.

The chemical structure and function of metabolites and the identifications from the Mummichog analysis were searched in the KEGG database (https://www.genome.jp/kegg/compound/) (*Kanehisa et al., 2014*), BiGG (http://bigg.ucsd.edu/universal/metabolites/) (*King et al., 2016*), the Edinburgh human metabolic network reconstruction (*Ma et al., 2007*) and the above-mentioned HMDB.

# RESULTS

## Body-Mass-Index (BMI) and body fat content of participants

Table 1 summarizes statistical data of the 153 participants. Of the 67 women and 86 men, 66 presented normal weight, 62 had overweight, and 25 were obese. Comparing female and male soldiers, the latter exhibited a higher prevalence of overweight and obesity. As expected, the groups with higher BMI also presented a higher body fat content, suggesting metabolic differences between these groups.

## Urinary metabolomics raw data processing and filtering

Figure 2 shows the number of features in the different sample groups and blank samples. We removed data sets of presumably empty samples and technical outliers by comparing the number of features with blank injections and eliminating all analyses with less than 4,000 features.

After clean-up, 52 samples of healthy, 47 overweight, and 21 obese individuals were left. We used these 120 data sets for further analysis. The healthy group showed 5,717 to 9,657, the overweight group 5,559 to 10,447, and the obese group 5,575 to 9,436 features.

## Identification of metabolic identities with MetaboAnalyst

First, we applied a cluster analysis with the sparse PLS-DA (sPLS-DA) algorithm (*Lê Cao, Boitard & Besse, 2011*), which indicates distinct metabolic identities of healthy, overweight, and obese individuals. However, the clustering is far from perfect, and especially the group of overweight individuals does not separate well from the other groups (Fig. 3A). We discussed the difficulty of clustering metabolic data in an earlier paper (*Winkler, 2015*).

**Table 1 General characteristics and anthropometric measurements of the soldiers by normal weight, overweight and obesity (Data are presented as mean ± SD).**

| n | Normal weight 66 | Overweight 62 | Obesity 25 | Global 153 |
|---|---|---|---|---|
| Age [years] | 27.74 ± 3.53 | 29.81 ± 4.53 | 37.83 ± 6.79 | 30.20 ± 5.73 |
| Age range | 22–45 | 22–45 | 29–49 | 22–49 |
| Gender | | | | |
| Female (% n) | 43 (28.1) | 18 (11.8) | 6 (3.9) | 67 (43.8) |
| Male (% n) | 23 (15.0) | 44 (28.8) | 19 (12.4) | 86 (56.2) |
| Weight [kg] | 61.05 ± 7.32 | 75.46 ± 6.18 | 84.02 ± 12.29 | 70.79 ± 11.77 |
| Height [m] | 1.62 ± 0.05 | 1.66 ± 0.06 | 1.60 ± 0.05 | 1.63 ± 0.06 |
| BMI [kg/m$^2$] | 23.02 ± 1.45 | 27.08 ± 1.33 | 33.33 ± 2.41 | 26.39 ± 3.88 |
| Body fat [%] | 25.09 ± 6.97 | 27.51 ± 6.28 | 34.63 ± 4.75 | 27.7. ± 7.10 |

**Notes.**
BMI, Body Mass Index.

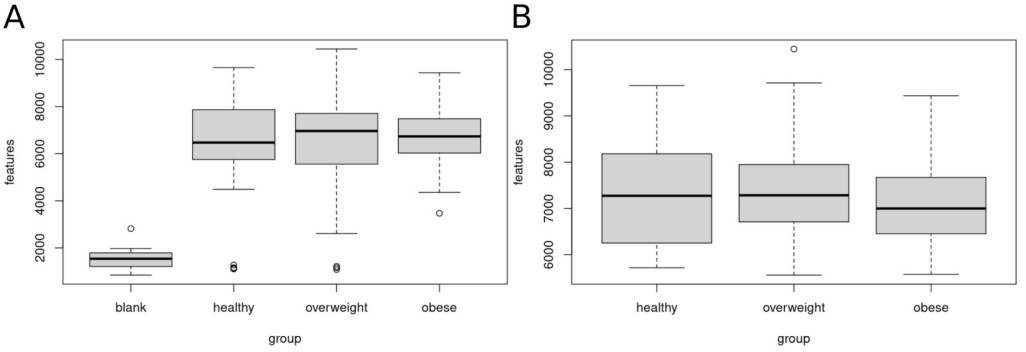

**Figure 2 Clean-up of raw data.** Sample data sets with less than 4,000 features were removed. (A) Boxplot of features (A) before clean-up, (B) after removal of samples with less than 4,000 features. A total of 120 data sets of healthy, overweight and obese individuals were used for further analyses.

To test if we could distinguish between healthy participants and others, we joined the overweight and obese groups and applied an orthogonal projection to latent structures data analysis (OPLS-DA) (*Trygg & Wold, 2002*). As a result, two clusters were separated reasonably well, (1) samples of healthy individuals and (2) samples of overweight and obese soldiers (Fig. 3B).

The classification is imperfect; however, the graphics represent the medical situation of clearly healthy, obviously sick, and patients in transition. Consequently, we can discriminate between two metabolic identities of normal-weight and overweight/obese soldiers.

## Statistical analysis of fold-changes

Using the same parameters for uploading the data (see 'Methods'), but only defining two groups, *i.e.,* healthy and obese-overweight, we created the Volcano plot shown in Fig. 4. We did this analysis in the one-factor statistical analysis module of MetaboAnalyst. We defined
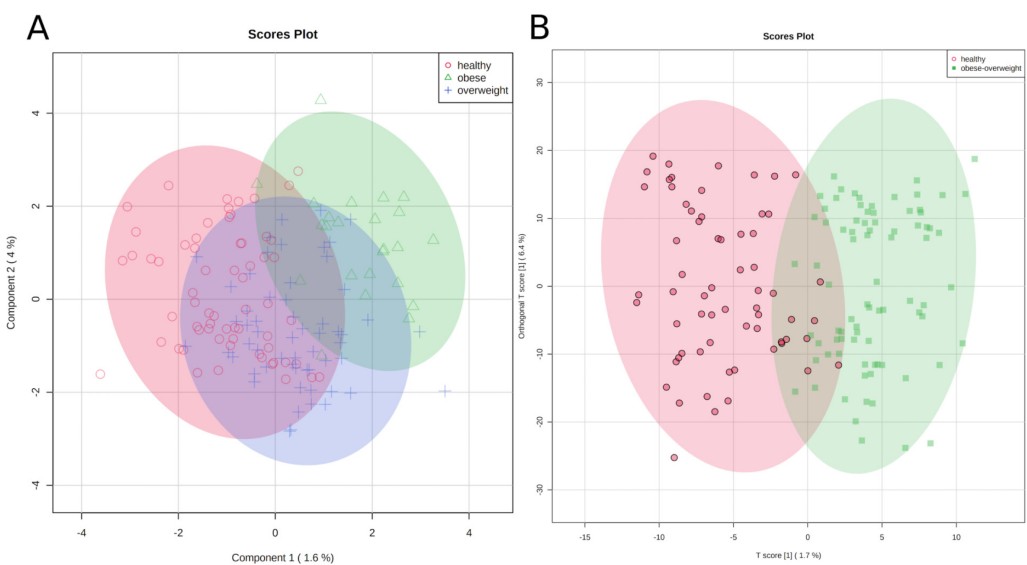

**Figure 3 Metabolic identity of healthy, overweight and obese groups.** (A) The clusters of sPLS-DA show overlapping of the three sample classes. The healthy and obese group can be more clearly discriminated, whereas the overweight group is located in between them. (B) OPLS-DA scores separate the samples of healthy individuals from overweight and obese soldiers.

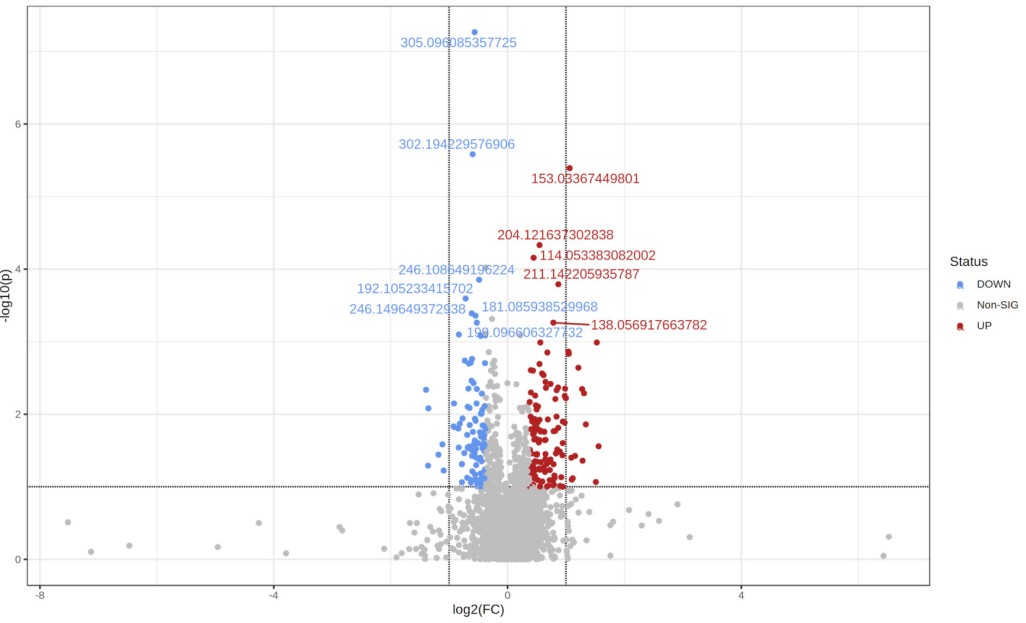

**Figure 4 The Volcano plot shows metabolic features with a *P*-value <0.1 and a fold-change of 1.3.**

non-parametric Wilcoxon rank-sum tests, a fold-change of 1.3 and a *p*-value threshold of 0.1 (raw), with equal group variance.

Two hundred twenty-five significant differential variables were detected and subjected to an Adaptive Boost data mining analysis.
**Table 2  Predictive classification model with the Adaptive Boost algorithm.**

| | | Predicted | | |
| | Actual | Healthy | Obese-overweight | Error [%] |
|---|---|---|---|---|
| Training | Healthy | 44 | 0 | 0.0 |
| | Obese-overweight | 0 | 58 | 0.0 |
| Validation | Healthy | 6 | 3 | 33.3 |
| | Obese-overweight | 2 | 10 | 16.7 |
| Testing | Healthy | 9 | 2 | 18.2 |
| | Obese-overweight | 1 | 11 | 8.3 |
| Overall | Healthy | 59 | 5 | 7.8 |
| | Obese-overweight | 3 | 79 | 3.7 |

## Adaptive boost analysis

The preselected 225 variables were loaded into R/Rattle (*Williams, 2009*; *Williams, 2011*) for further evaluation and split into three partitions for training, validation, and testing (70/15/15). Variables with missing values were deleted. The following parameters were used:

```
ada(Group ~ ., data = crs$dataset[crs$train, c(crs$input, crs$target)],
    control = rpart::rpart.control(maxdepth = 6, cp = 0.01, minsplit = 20,
        xval = 10), iter = 500)
```

Table 2 summarizes the results of the model building process. The overall error of the model is 5.5%, with an average class error of 5.75%.

Consequently, the classification between healthy and obese-overweight persons based on urinary metabolomics profiles is highly reliable, considering natural variations.

The important variables that contribute most to correct classification are shown in Fig. 5.

## Biomarker analysis

Table 3 lists important variables from the Ada Boost analysis with at least a 1.3-fold significant change. Those ions are possible biomarkers for weight-related metabolic studies.

## Mummichog analysis: metabolic pathway enrichment

To explore affected metabolic pathways and facilitate the identification of metabolites, we performed a Mummichog analysis in MetaboAnalyst (see 'Methods').

As indicated in Table 4 and Fig. 6, five pathways demonstrated enrichment above the defined threshold limits:

- Urea cycle/amino group metabolism
- Alanine and aspartate metabolism
- Drug metabolism—cytochrome P450
- Aspartate and asparagine metabolism
- Ubiquinone biosynthesis.

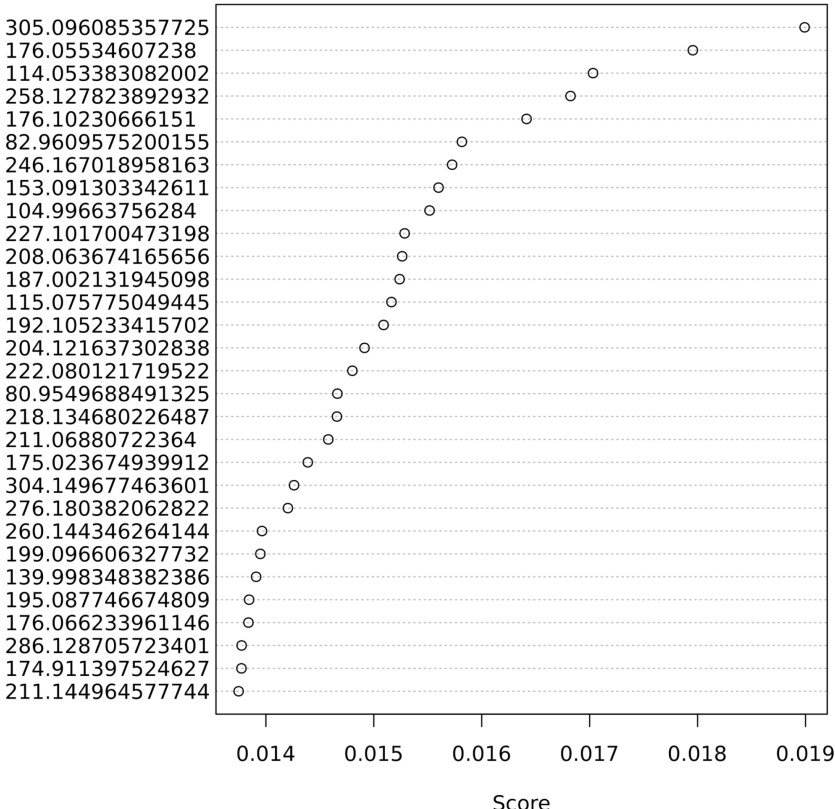

**Figure 5  Variable importance for the predictive Adaptive Boost classification model.**

Especially the appearance of urea cycle/amino group metabolism as the first hit gives confidence to the Mummichog algorithm since no information about the origin of the samples was given to the MetaboAnalyst platform.

Thus, ions assigned to metabolites of enriched pathways have increased confidence in our further discussion.

## DISCUSSION

### Classification of normal weight *vs.* overweight-obese, based on metabolic signature

To develop a predictive classification model, we used the untargeted LC-MS features with at least a 1.3-fold change. The features correspond to ions with a particular retention time. Although a 30% increased or decreased metabolite level might not be critical for health, it can indicate a disturbed pathway.

Identifying compounds corresponding to the features is theoretically possible. However, the reliable assignment of metabolites is tedious (*Rathahao-Paris et al., 2015*; *Jeffryes et al., 2015*; *Fuente et al., 2019*; *Djoumbou-Feunang et al., 2019*; *Dührkop et al., 2019*), and the

**Table 3  Important variables from the Ada Boost analysis with at least 1.3-fold significant change.**

| Ada Boost | m/z | FC | log2 (FC) | raw.pval | −log10 (p) |
|---|---|---|---|---|---|
| 1 | 305.096085357725 | 0.67706 | −0.56264 | 0.000000054252 | 7.2656 |
| 2 | 176.05534607238 | 0.76713 | −0.38246 | 0.00081848 | 3.087 |
| 3 | 114.053383082002 | 1.3627 | 0.44649 | 0.000069642 | 4.1571 |
| 4 | 258.127823892932 | 1.4759 | 0.56159 | 0.0010258 | 2.989 |
| 5 | 176.10230666151 | 1.3729 | 0.45718 | 0.022281 | 1.6521 |
| 6 | 82.9609575200155 | 0.68689 | −0.54184 | 0.039329 | 1.4053 |
| 7 | 246.167018958163 | 1.566 | 0.64711 | 0.041643 | 1.3805 |
| 8 | 153.091303342611 | 1.4299 | 0.51588 | 0.012894 | 1.8896 |
| 9 | 104.99663756284 | 0.75266 | −0.40993 | 0.014395 | 1.8418 |
| 10 | 227.101700473198 | 1.968 | 0.97672 | 0.013038 | 1.8848 |
| 11 | 208.063674165656 | 1.4688 | 0.55469 | 0.098829 | 1.0051 |
| 12 | 187.002131945098 | 0.75863 | −0.39852 | 0.032069 | 1.4939 |
| 13 | 115.075775049445 | 0.6563 | −0.60758 | 0.0017274 | 2.7626 |
| 14 | 192.105233415702 | 0.60822 | −0.71733 | 0.00025415 | 3.5949 |
| 15 | 204.121253887635 | 1.924 | 0.94407 | 0.099638 | 1.0016 |
| 16 | 222.080121719522 | 1.788 | 0.83835 | 0.010779 | 1.9674 |
| 17 | 80.9549688491325 | 0.70797 | −0.49824 | 0.04125 | 1.3846 |
| 18 | 218.134680226487 | 2.1311 | 1.0916 | 0.039707 | 1.4011 |
| 19 | 211.06880722364 | 1.3152 | 0.39528 | 0.010779 | 1.9674 |
| 20 | 175.023674939912 | 0.75944 | −0.39698 | 0.094865 | 1.0229 |
| 21 | 304.149677463601 | 1.3526 | 0.43569 | 0.0025023 | 2.6017 |
| 22 | 276.180382062822 | 0.58665 | −0.76942 | 0.011404 | 1.9429 |
| 23 | 260.144346264144 | 1.7745 | 0.82742 | 0.034686 | 1.4598 |
| 24 | 199.096606327732 | 0.69475 | −0.52543 | 0.00054643 | 3.2625 |
| 25 | 139.998348382386 | 0.68953 | −0.53631 | 0.050208 | 1.2992 |
| 26 | 195.087746674809 | 1.7269 | 0.78819 | 0.017119 | 1.7665 |
| 27 | 176.066233961146 | 0.72685 | −0.46027 | 0.00081848 | 3.087 |
| 28 | 286.128705723401 | 1.388 | 0.47301 | 0.0055271 | 2.2575 |
| 29 | 174.911397524627 | 1.4127 | 0.49845 | 0.0085721 | 2.0669 |
| 30 | 211.144964577744 | 1.322 | 0.40276 | 0.016049 | 1.7946 |

Notes.

Ada Boost, Ada Boost rank; m/z, mass-to-charge ratio of feature; FC, fold-change; pval, p-value.

data mining models are helpful without knowing the related compounds (*Winkler, 2015*). Thus, we limited the identification of compounds to important variables.

The OPLS-DA analysis already indicated distinct metabolic identities (Fig. 3B) for normal weight and overweight-obese individuals. A predictive model that we developed with the Adaptive Boost algorithm was able to classify normal weight and overweight-obese individuals with an overall error of 5.5% (Table 2). Notably, the highest errors were found in the validation and testing data of healthy soldiers wrongly classified as overweight or obese. These assignments could indicate a possible tendency of the soldiers to gain weight.

The Adaptive Boost model demonstrates metabolic differences between normal weight and overweight-obese individuals, which can be used for classification. Further, the

Albores-Mendez et al. (2022), *PeerJ*, DOI 10.7717/peerj.13754

**Table 4 Enriched pathways from the Mummichog analysis.**

| Pathway | Pathway tot. | Hits tot. | Hits sig. | Expected | FET | EASE | Gamma | Emp. Hits | Emp. | Pathway No. | Cpd. Hits |
|---|---|---|---|---|---|---|---|---|---|---|---|
| Urea cycle/amino group metabolism | 85 | 50 | 10 | 3.7797 | 0.0045702 | 0.0136 | 0.039704 | 0 | 0 | P1 | C00062; C04441; C04692; C00437; C00073; C00019; C00242; C01449; C01250; C00547; C00049 |
| Alanine and Aspartate Metabolism | 30 | 20 | 5 | 1.334 | 0.016982 | 0.065906 | 0.041654 | 0 | 0 | P2 | C00062; C00940; C01042; C00402; C00049 |
| Drug metabolism - cytochrome P450 | 53 | 48 | 7 | 2.3567 | 0.079575 | 0.17018 | 0.046002 | 0 | 0 | P3 | C16582; C16604; C16550; C07501; C16609; C16584; C16586 |
| Aspartate and asparagine metabolism | 114 | 77 | 9 | 5.0692 | 0.14967 | 0.25437 | 0.050052 | 0 | 0 | P4 | C00437; C01239; CE1938; C00402; C05932; C00062; C02571; C04540; C03078; C03415; CE1943; C00049 |
| Lysine metabolism | 52 | 28 | 4 | 2.3123 | 0.17608 | 0.38004 | 0.057276 | 0 | 0 | P5 | C00019; C06157; C03793; C01259 |
| Ubiquinone Biosynthesis | 10 | 7 | 2 | 0.44467 | 0.10051 | 0.43686 | 0.061142 | 0 | 0 | P6 | C01179; C00019 |
| Vitamin B3 (nicotinate and nicotinamide) metabolism | 28 | 19 | 3 | 1.2451 | 0.18615 | 0.44767 | 0.061929 | 0 | 0 | P7 | C00062; C00019; C00049 |
| Vitamin B1 (thiamin) metabolism | 20 | 9 | 2 | 0.88933 | 0.15545 | 0.5223 | 0.067899 | 0 | 0 | P8 | C06157; C16255 |
| Tyrosine metabolism | 160 | 103 | 9 | 7.1147 | 0.43083 | 0.57147 | 0.072443 | 0 | 0 | P9 | C05350; C00019; C05852; C03758; C02505; C00547; CE5547; C00642; C00082; C05576; C07453; C00355; C01179; C00268; C05584; C05587; C05588; C04043; CE2174; CE2176; CE2173 |
| Arginine and Proline Metabolism | 45 | 38 | 4 | 2.001 | 0.35481 | 0.58556 | 0.073852 | 0 | 0 | P10 | C00062; C00073; C00019; C00049; C05933 |
| Biopterin metabolism | 22 | 14 | 2 | 0.97827 | 0.3058 | 0.68367 | 0.085412 | 2 | 0.02 | P11 | C04244; C00268; C00082 |
| Pyrimidine metabolism | 70 | 45 | 4 | 3.1127 | 0.48368 | 0.70125 | 0.08789 | 0 | 0 | P12 | C00214; C00881; C00475; C00049 |

Albores-Mendez et al. (2022), *PeerJ*, DOI 10.7717/peerj.13754

**Table 4** (*continued*)

| Pathway | Pathway tot. | Hits tot. | Hits sig. | Expected | FET | EASE | Gamma | Emp. Hits | Emp. | Pathway No. | Cpd. Hits |
|---|---|---|---|---|---|---|---|---|---|---|---|
| Tryptophan metabolism | 94 | 74 | 6 | 4.1799 | 0.54076 | 0.70613 | 0.088605 | 0 | 0 | P13 | C05647; C00019; C05651; C02220; C00078; C00268; C00328; C04409; C03227; C00525 |
| Starch and Sucrose Metabolism | 33 | 15 | 2 | 1.4674 | 0.33598 | 0.70875 | 0.088995 | 0 | 0 | P14 | CE2837; C01083; C00208 |
| Vitamin B9 (folate) metabolism | 33 | 16 | 2 | 1.4674 | 0.36578 | 0.73186 | 0.092598 | 0 | 0 | P15 | C01045; C00504 |
| Butanoate metabolism | 34 | 20 | 2 | 1.5119 | 0.47883 | 0.80744 | 0.10716 | 1 | 0.01 | P16 | C05548; C02727 |
| Porphyrin metabolism | 43 | 20 | 2 | 1.9121 | 0.47883 | 0.80744 | 0.10716 | 0 | 0 | P17 | C05520; C00931 |
| Xenobiotics metabolism | 110 | 59 | 4 | 4.8913 | 0.7018 | 0.8572 | 0.1204 | 0 | 0 | P18 | C00870; C14853; C06205; C14871 |
| Histidine metabolism | 33 | 25 | 2 | 1.4674 | 0.60163 | 0.87285 | 0.12555 | 8 | 0.08 | P19 | C00439; C00019 |
| Methionine and cysteine metabolism | 94 | 47 | 3 | 4.1799 | 0.73432 | 0.89655 | 0.13469 | 0 | 0 | P20 | C08276; C00019; C00073 |
| Sialic acid metabolism | 107 | 28 | 2 | 4.7579 | 0.66429 | 0.90095 | 0.13661 | 0 | 0 | P21 | C00140; C00645; C00243 |
| Purine metabolism | 80 | 53 | 3 | 3.5573 | 0.80598 | 0.93105 | 0.15258 | 0 | 0 | P22 | C00499; C00242; C00049 |
| Galactose metabolism | 41 | 34 | 2 | 1.8231 | 0.7658 | 0.93997 | 0.15864 | 0 | 0 | P23 | C00140; C05400; C05402; C05399; C00243; C00089 |
| Glycine, serine, alanine and threonine metabolism | 88 | 60 | 3 | 3.9131 | 0.86848 | 0.95761 | 0.17378 | 1 | 0.01 | P24 | C00062; C00019; C00073 |
| Androgen and estrogen biosynthesis and metabolism | 95 | 71 | 3 | 4.2243 | 0.93142 | 0.98074 | 0.20732 | 0 | 0 | P25 | C02538; C05293; C00019; C03917; C04373; C04295; C00523 |
| Glycerophospholipid metabolism | 156 | 49 | 2 | 6.9368 | 0.9118 | 0.98298 | 0.21248 | 1 | 0.01 | P26 | C00019; C00670 |
| Leukotriene metabolism | 92 | 54 | 2 | 4.0909 | 0.93745 | 0.98885 | 0.22988 | 0 | 0 | P27 | C03577; CE5140; CE4995 |

Albores-Mendez et al. (2022), *PeerJ*, DOI 10.7717/peerj.13754

**Table 4** (*continued*)

| Pathway | Pathway tot. | Hits tot. | Hits sig. | Expected | FET | EASE | Gamma | Emp. Hits | Emp. | Pathway No. | Cpd. Hits |
|---|---|---|---|---|---|---|---|---|---|---|---|
| C21-steroid hormone biosynthesis and metabolism | 112 | 81 | 2 | 4.9803 | 0.99121 | 0.99889 | 0.31857 | 0 | 0 | P28 | C03917; C02538; C04373; C00523 |
| Hyaluronan Metabolism | 8 | 4 | 1 | 0.35573 | 0.28138 | 1 | 1 | 0 | 0 | P29 | C00140 |
| Glycolysis and Gluconeogenesis | 49 | 32 | 1 | 2.1789 | 0.93051 | 1 | 1 | 0 | 0 | P30 | C01136 |
| Hexose phosphorylation | 20 | 16 | 1 | 0.88933 | 0.73463 | 1 | 1 | 2 | 0.02 | P31 | C01083; C00089 |
| Keratan sulfate degradation | 68 | 6 | 1 | 3.0237 | 0.391 | 1 | 1 | 0 | 0 | P32 | C00140 |
| Carnitine shuttle | 72 | 23 | 1 | 3.2016 | 0.8521 | 1 | 1 | 0 | 0 | P33 | pcrn |
| Alkaloid biosynthesis II | 10 | 6 | 1 | 0.44467 | 0.391 | 1 | 1 | 0 | 0 | P34 | egme |
| Parathio degradation | 6 | 5 | 1 | 0.2668 | 0.33844 | 1 | 1 | 0 | 0 | P35 | C00870 |
| Electron transport chain | 7 | 3 | 1 | 0.31127 | 0.21943 | 1 | 1 | 0 | 0 | P36 | C00390 |
| Vitamin H (biotin) metabolism | 5 | 5 | 1 | 0.22233 | 0.33844 | 1 | 1 | 0 | 0 | P37 | C00120 |
| De novo fatty acid biosynthesis | 106 | 22 | 1 | 4.7135 | 0.83919 | 1 | 1 | 0 | 0 | P38 | C06429 |
| Vitamin A (retinol) metabolism | 67 | 41 | 1 | 2.9793 | 0.96749 | 1 | 1 | 0 | 0 | P39 | C16679; C16677; C16680 |
| Valine, leucine and isoleucine degradation | 65 | 26 | 1 | 2.8903 | 0.88497 | 1 | 1 | 14 | 0.14 | P40 | C00123; C00407 |
| Fatty Acid Metabolism | 63 | 15 | 1 | 2.8014 | 0.71158 | 1 | 1 | 0 | 0 | P41 | C02571 |
| Heparan sulfate degradation | 34 | 5 | 1 | 1.5119 | 0.33844 | 1 | 1 | 0 | 0 | P42 | C00140 |
| TCA cycle | 31 | 18 | 1 | 1.3785 | 0.77539 | 1 | 1 | 0 | 0 | P43 | C00390 |
| Arachidonic acid metabolism | 95 | 75 | 1 | 4.2243 | 0.99823 | 1 | 1 | 0 | 0 | P44 | C04741; C04843; C14782; C14814; C00639 |
| Phosphatidylinositol phosphate metabolism | 59 | 29 | 1 | 2.6235 | 0.91057 | 1 | 1 | 0 | 0 | P45 | C01235 |

Albores-Mendez et al. (2022), *PeerJ*, DOI 10.7717/peerj.13754

**Table 4** (*continued*)

| Pathway | Pathway tot. | Hits tot. | Hits sig. | Expected | FET | EASE | Gamma | Emp. Hits | Emp. | Pathway No. | Cpd. Hits |
|---|---|---|---|---|---|---|---|---|---|---|---|
| Prostaglandin formation from arachidonate | 78 | 61 | 1 | 3.4684 | 0.99409 | 1 | 1 | 0 | 0 | P46 | C04741; C05959; C00639 |
| Vitamin B6 (pyridoxine) metabolism | 11 | 8 | 1 | 0.48913 | 0.48401 | 1 | 1 | 3 | 0.03 | P47 | C00314 |
| N-Glycan Degra-dation | 16 | 8 | 1 | 0.71147 | 0.48401 | 1 | 1 | 1 | 0.01 | P48 | C00140 |
| Vitamin B12 (cyanocobalamin) metabolism | 9 | 3 | 1 | 0.4002 | 0.21943 | 1 | 1 | 0 | 0 | P49 | C00019 |
| Carbon fixation | 10 | 10 | 1 | 0.44467 | 0.5629 | 1 | 1 | 0 | 0 | P50 | C00049 |
| Nitrogen metabolism | 6 | 4 | 1 | 0.2668 | 0.28138 | 1 | 1 | 4 | 0.04 | P51 | C00049 |
| Drug metabolism - other enzymes | 31 | 22 | 1 | 1.3785 | 0.83919 | 1 | 1 | 5 | 0.05 | P52 | C16631 |
| Aminosugars metabolism | 69 | 25 | 1 | 3.0682 | 0.87491 | 1 | 1 | 3 | 0.03 | P53 | C00140; C00645 |
| Beta-Alanine metabolism | 20 | 15 | 1 | 0.88933 | 0.71158 | 1 | 1 | 11 | 0.11 | P54 | C00049 |
| Prostaglandin formation from dihomo gama-linoleic acid | 11 | 8 | 1 | 0.48913 | 0.48401 | 1 | 1 | 0 | 0 | P55 | C04741 |

**Notes.**

Pathway tot., total number of compounds in this pathway; Hits tot., total of putative hits for this pathway; Hits sig., significant hits; Expected, randomly expected hits; FET, Fisher's exact test; EASE, adjusted FET; Gamma, gamma corrected *p*-value; Emp., empirical compounds, such as adducts; Cpd., compound (with KEGG database identifier).

The compounds corresponding to the database identifiers are provided as a Table S1.

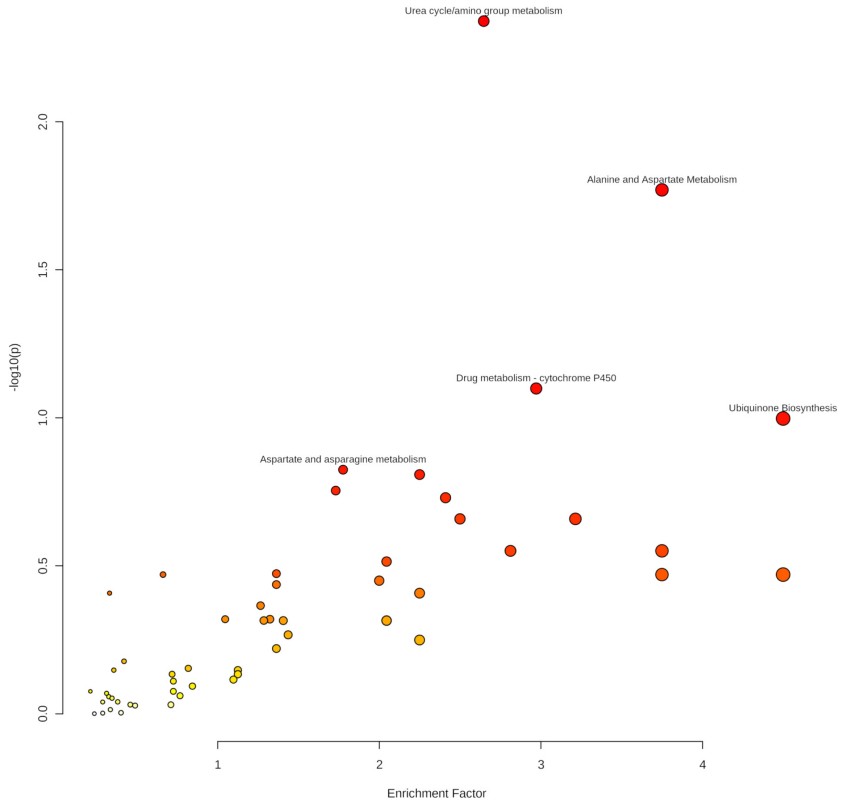

**Figure 6  Enriched pathways from the Mummichog analysis.**

Adaptive Boost could provide a sensitive method to estimate the metabolic state and the tendency of a person to gain weight. However, additional studies are necessary to evaluate the performance of Adaptive Boost models with untargeted metabolic data as a predictive tool in clinical diagnostics and treatment.

## Metabolic pathways in obesity-overweight and potential biomarkers

Compiling the biomarker candidate ions with likely metabolite identifications resulted in Fig. 7.

Several ions and the metabolic pathway integration-derived metabolites hint at S-adenosyl-L-methionine (SAM). A previous study reported a 42% increase of SAM in the serum of test persons who were overfed by 1,250 kcal per day and gained weight above the median (*Elshorbagy et al., 2016*). SAM is synthesized from methionine and ATP and is a key metabolite since it donates methyl groups to different molecules, such as DNA, RNA, proteins, and lipids, in enzymatic reactions. The demethylated S-adenosyl-homocysteine (SAH) is hydroxylated by adenosylhomocysteinase, resulting in adenosine and homocysteine. Methionine synthase builds methionine by transferring a methyl group from 5-methyl-tetrahydrofolate to homocysteine (*Finkelstein, 2000*).

Several of these reactions have been reported to be altered in obesity. For example, high serum levels of homocysteine have been correlated with reduced high-density lipoprotein

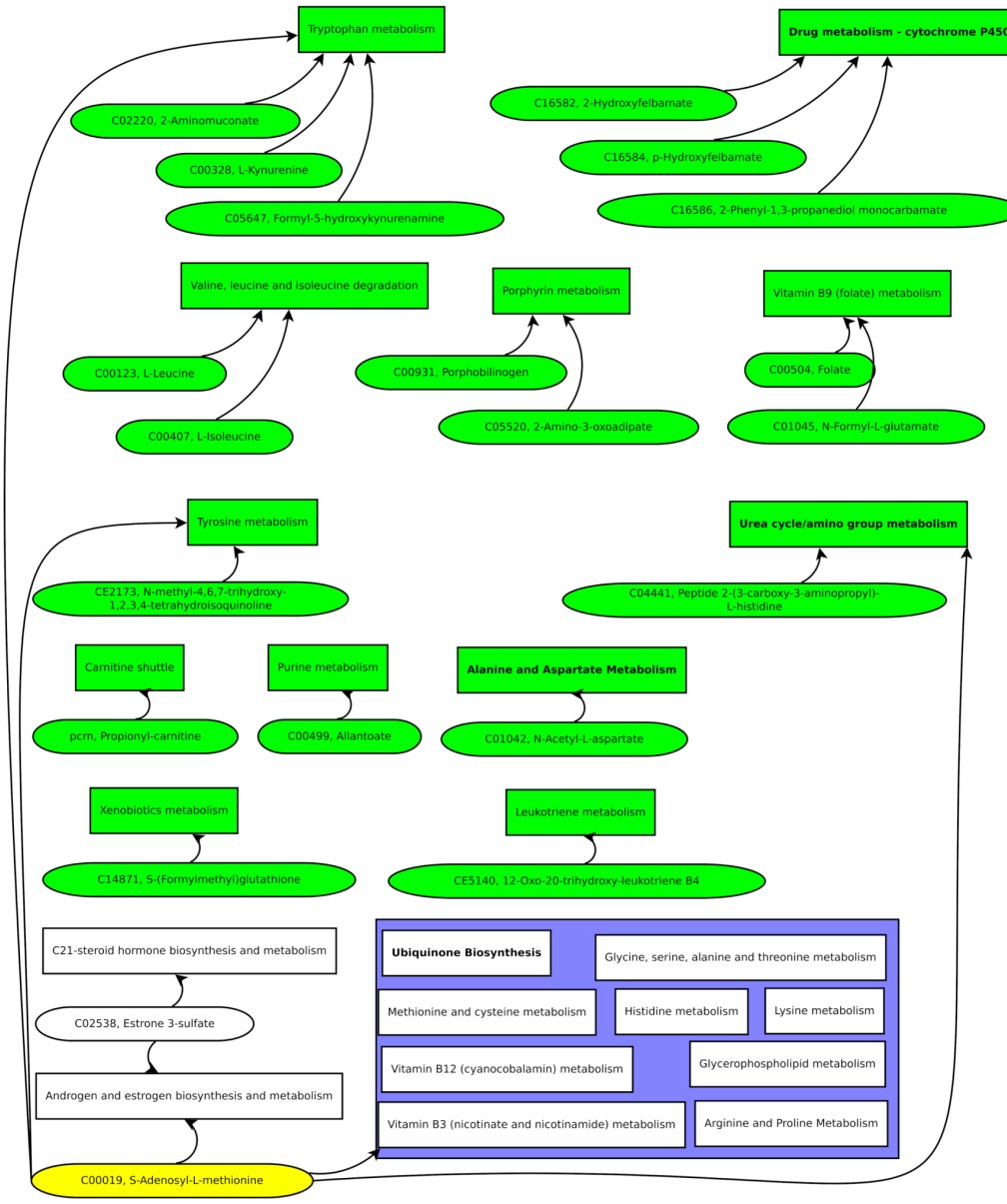

**Figure 7  Green pathways contain at least one unique putative compound.** Green putative compounds are unique for one pathway.

(HDL) levels. The accumulation of homocysteine comes with lower SAM and SAH levels, leading to a diminished production of phosphatidylcholine, which is essential for the production of low-density lipoproteins (LDL) and very-low-density lipoproteins (VLDL) (*Obeid & Herrmann, 2009*). Hyperlipidemia with increased serum homocysteine increases the risk of developing an atherosclerotic disease in overweight patients (*Glueck et al., 1995*). In addition, elevated serum homocysteine is related to hepatic steatosis. The later effect was pronounced with low folate intake (*Gulsen et al., 2005*). Strikingly, we also found the folate metabolism affected in our present study.

Another altered SAM-related pathway, we detected, is related to nicotinamide metabolism. Nicotinamide-N-methyl transferase (NNMT) methylates nicotinamide, using SAM as a methyl donor (*Ramsden et al., 2017*). As a result, NNMT is enriched in adipose tissue and the liver of patients with obesity and type 2 diabetes mellitus (DM2) (*Kraus et al., 2014*).

The possibility of detecting excess food energy intake in urine by measuring SAM would provide a non-invasive method for monitoring patients during weight-loss diets and professionals who require high physical fitness, such as soldiers. Thus, the level of SAM will be assayed in the following study during the treatment of obese military personnel.

In addition, several ions that putatively correspond to compounds from amino acid metabolism were identified. Changes in amino acid levels and related metabolites in obese patients have been reported in several studies (*Xie, Waters & Schirra, 2012*; *Maltais-Payette et al., 2018*; *Yu et al., 2018*). Therefore, our finding is expectable. However, since we found the alteration of amino acid pathways through a variable importance analysis of untargeted metabolomics data, we suggest a high relevance of amino acid-related biomarkers compared to other groups of compounds such as TCA-cycle metabolites.

Therefore, besides the SAM level, we will investigate the role of amino acid metabolism in obesity and weight reduction in future studies.

## CONCLUSIONS

An Ada Boost model based on urinary metabolomics data could discriminate obese and overweight from healthy military personnel with a low overall error rate of 5.5%, indicating a metabolic signature related to the excessive ingestion of food.

Important variables from data mining, statistical analyses, and metabolic pathway enrichment analysis suggest S-adenosyl-methionine (SAM) as a possible urine biomarker for overfeeding. Increased SAM levels were found for overfed people in plasma, but monitoring SAM in urine could be used daily for close follow-up of patients, for example, in the treatment of losing weight or persons that need a high level of physical fitness, such as soldiers.

As well, the amino acid metabolism showed significant changes.

Therefore, in ongoing studies, we include SAM, amino acid metabolism compounds, and acylcarnitines for evaluating the metabolic state of military personnel. In the future, our results will support the design of low-cost biochemical assays for the broad public.

## ACKNOWLEDGEMENTS

We thank the Military Graduate School of Health (E.MG.S.), and CINVESTAV Irapuato for all the support and facilities provided for the materialization of the project.

### Funding

This project was supported by The Budget Program A022, Military Research and Development in Coordination with Public Universities, Public Higher Education Institutions, and/or other Public Research Centers and the Secretary of National Defense, Mexico. The funders had no role in study design, data collection and analysis, decision to publish, or preparation of the manuscript.

### Grant Disclosures

The following grant information was disclosed by the authors:
The Budget Program A022, Military Research and Development in Coordination with Public Universities, Public Higher Education Institutions.
Public Research Centers and the Secretary of National Defense, Mexico.

### Competing Interests

Robert Winkler is an Academic Editor of PeerJ and Section Editor of PeerJ Plant Biology.

### Author Contributions

- Exsal M. Albores-Mendez conceived and designed the experiments, performed the experiments, analyzed the data, prepared figures and/or tables, authored or reviewed drafts of the article, and approved the final draft.
- Alexis D. Aguilera Hernández conceived and designed the experiments, performed the experiments, authored or reviewed drafts of the article, and approved the final draft.
- Alejandra Melo-González performed the experiments, authored or reviewed drafts of the article, and approved the final draft.
- Marco A. Vargas-Hernández conceived and designed the experiments, authored or reviewed drafts of the article, and approved the final draft.
- Neptalí Gutierrez de la Cruz performed the experiments, authored or reviewed drafts of the article, and approved the final draft.
- Miguel A. Vazquez-Guzman conceived and designed the experiments, authored or reviewed drafts of the article, and approved the final draft.
- Melchor Castro-Marín performed the experiments, authored or reviewed drafts of the article, and approved the final draft.
- Pablo Romero-Morelos performed the experiments, authored or reviewed drafts of the article, and approved the final draft.
- Robert Winkler analyzed the data, prepared figures and/or tables, authored or reviewed drafts of the article, and approved the final draft.

### Human Ethics

The following information was supplied relating to ethical approvals (*i.e.*, approving body and any reference numbers):

This work was approved by the Research Committee and the Bioethics Committee of the Escuela Militar de Medicina, Universidad del Ejército y Fuerza Aérea Mexicanos (reg. 0129012020).

## Data Availability

The mass spectrometry data in .mzML format, KNIME workflow for raw data processing, and data matrices used for MetaboAnalyst analyses are available at Zenodo:

Winkler Robert. (2022). SUPEREGO urinary metabolomics [Data set]. Zenodo. https://doi.org/10.5281/zenodo.6091674.

## Supplemental Information

Supplemental information for this article can be found online at http://dx.doi.org/10.7717/peerj.13754#supplemental-information.

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
