# Peer review of "A diagnostic model for overweight and obesity from untargeted urine metabolomics of soldiers"

_PeerJ, doi:10.7717/peerj.13754_

## Round 0.1 · original submission · Major Revisions

Reviewers have identified some methodological problems, such as not taking into account some potential confounding variables. This should be clarified in the revised version of the manuscript.

Reviewer 1 ·

Basic reporting

The manuscript was developed in the way weak readable (descriptions, tables, figures). The manuscript contains methodological errors. There is lack of the aim of the study (Introduction, line 76-79, it is very difficult to consider as the aim of the study). There is no relations between: a title – aims – conclusions.

Experimental design

General description of the research. There is no information, e.g. when (time of day) the tests were performed. The protocol of data acquisition, procedures, investigated parameters, methods of measurements and apparatus should be described in sufficient detail to allow other scientists to reproduce the results. It wasn't written about application value of research. What is this method better than the currently used methods of assessing overweight and obesity? Why this research were conducted among soldiers?

Validity of the findings

It was not substantiated why the search for a diagnostic model for overweight and obesity from unterpeted urine metabolics of soldiers was undertaken. Why this research were conducted among soldiers?
The sentences written in the Conclusions section cannot be considered as conclusions in the context of the conducted research. Conclusions should be linked with the goals of the study. Unqualified statements and conclusions not completely supported by the obtained data should be avoided.

Additional comments

Taking into account methodological errors, I do not recommend the manuscript for publication.

Reviewer 2 ·

Basic reporting

The current manuscript is well written and after some minor modifications it will be published.

Experimental design

Experimental design is looking fine.

Validity of the findings

Results are fine in current form but need some extra parameters related to subject.

Additional comments

In the current study “A diagnostic model for overweight and obesity from untargeted urine metabolomics of soldiers. Definitely it is good study which highlighted main concern related to obesity in soldiers. But there are some major points to be added to improve the quality of manuscript in result and discussion section. As authors discussed regarding BMI in the current manuscript, waist hip ratio should be included with lipid profile which will provide the strength of future Science community and soldiers.
There is some typo and grammatical mistake should be rectifying in the revised version of the manuscript.
Overall, manuscript is well written and certainly provides strength to Scientist in coming research and boost up soldiers.

Annotated reviews are not available for download in order to protect the identity of reviewers who chose to remain anonymous.

Reviewer 3 ·

Basic reporting

The researcher used untargeted metabolomics for 146 individuals stratified by BMI levels. The main objective analysis was to analyze metabolic expression in urine by BMI groups. In general, at this stage of the knowledge, we know a lot about amino acid serum abnormalities, but less about urinary metabolomics.

Experimental design

Strengths of the study. The researchers assembled a metabolomic study in urine, with an acceptable sample size of individuals. The results of the study confirm other studies reporting aminoacid as an early abnormal trait associated with obesity. The researchers used contemporary analysis for connecting metabolic pathways and made statistical analysis with the MetaboAnalyst software.
Weakness of the study. The researchers did not attempt to correct the confounders (at least is not clear how they approached decreasing the effect of age and sex).

Validity of the findings

Specific suggestions.
1. The BMI formula is incorrect (formula 1 on page 4/29), it should be weight in kilograms divided in height squared.
2. The MetaboAnalyst report (described in the methods section) can be included in the supplemental section, but the way how it is presented does not give useful information.
3. Please, justify the reason researchers used normalization by median values and after this, they re-transformed with square roots (double steps transformation). A better efficient approach is to use other well-known methods easier to interpret and with only one step transformation (i.e.: inverse normalization with ranks, a common way to normalize –omics).
4. How do they deal with extreme values once transformed with median and IQR?
5. To avoid confusion with the sample used and avoid contradictions in the number of individuals, they should describe the flux of the sample. For example, table 1 described 153 individuals but they analyzed 146.
6. They have to deal with confounders. Some mean values by BMI classification had very different distributions, for example, significant differences by age (normal weight has a mean value 10 years younger than people with obesity) and gender (normal weight was represented 1.9 times larger by women, meanwhile obesity 3.2 times larger frequency by men). How do the researchers analyze these confounders?
7. The volcano plot and adaptive boots analysis maintained the effect of the same confounders.
8. Figures 4 and 5, and table 3 should have names of metabolites in order of predictive importance, instead of the label of mz.
9. The effect of 1.3 fold change is very small. Is clinically worth it?
10. The called biomarker SAM is expressing/representing what clinical trait? In other words, what represents the amino acid metabolism differences in these groups. It is already known obesity can be measured by anthropometry, therefore, do the findings on urine metabolic differences are related to insulin resistance, inflammation, and early stages of insulin derangements, among others?

Additional comments

The study is original, engaging, and worth to be improved.

---

## Round 0.2 · Minor Revisions

Several aspects need to be clarified:

In the tables, compounds are referred to only as KEGG labels, rather than their common names. This makes it cumbersome for the reader to get insight from the paper, since he/she will have to run every reported KEGG number through KEGG to ascertain their identity. I suggest that authors provide a version of this table with full names of the compounds (either in the text or as SI).

Also, the legends to table 4 and table 3 do not properly convey what each column is supposed to represent: what is FC in table 3? What are Pathway tot. / Hits tot. / Hits sig./ FET /EASE/ Gamma/ Emp. Hits/etc . in table 4, and why is the number of compounds in each class sometimes different from the number of Hits sig.?

Reviewer 2 ·

Basic reporting

Manuscript is in publishable form now.

Experimental design

Good

Validity of the findings

Meaningful

Additional comments

No need for additional comments.

---

## Round 0.3 · accepted · Accept

Thank you for accepting the suggested changes and congratulations for the nice paper!